# Is There Such a Thing as a Genuine Cancer Stem Cell Marker? Perspectives from the Gut, the Brain and the Dental Pulp

**DOI:** 10.3390/biology9120426

**Published:** 2020-11-27

**Authors:** Crende Olatz, García-Gallastegui Patricia, Luzuriaga Jon, Badiola Iker, de la Hoz Carmen, Unda Fernando, Ibarretxe Gaskon, Pineda Jose Ramon

**Affiliations:** 1Department of Cell Biology and Histology, Faculty of Medicine and Nursing, University of the Basque Country (UPV/EHU), 48940 Leioa, Spain; olatz.crende@ehu.eus (C.O.); patricia.garcia@ehu.eus (G.-G.P.); jon.luzuriaga@ehu.eus (L.J.); iker.badiola@ehu.eus (B.I.); carmen.delahoz@ehu.eus (d.l.H.C.); fernando.unda@ehu.eus (U.F.); 2Achucarro Basque Center for Neuroscience Fundazioa, 48940 Leioa, Spain

**Keywords:** stem cells, cancer stem cells, dental pulp stem cells, glioma, colorectal cancer, cell markers, telomerase, alternative lengthening of telomeres, pluripotency core factors

## Abstract

**Simple Summary:**

Did you ever wonder why some tissues can produce very aggressive types of cancer whereas others are apparently immune to this devastating disease? One of the most accepted theories in the scientific community states that tumors are fueled by small numbers of key master cells called cancer stem cells, which mediate tumor relapse and metastasis. Much effort has been made to identify these cells by the characterization of their defining markers, in an attempt to eliminate these cells selectively. However, many of these markers are also present in other healthy stem cells in the body, including those found in some tissues like the dental pulp, which is known to be highly resistant to carcinogenesis. This brings up the question of whether there is indeed a genuine marker that can be used to unequivocally identify cancer stem cells. We set out to address this question by a systematic comparison of healthy stem cells and cancer stem cells of different body locations, and we discuss some key factors that play a role in the resistance of certain types of stem cells to malignant transformation.

**Abstract:**

The conversion of healthy stem cells into cancer stem cells (CSCs) is believed to underlie tumor relapse after surgical removal and fuel tumor growth and invasiveness. CSCs often arise from the malignant transformation of resident multipotent stem cells, which are present in most human tissues. Some organs, such as the gut and the brain, can give rise to very aggressive types of cancers, contrary to the dental pulp, which is a tissue with a very remarkable resistance to oncogenesis. In this review, we focus on the similarities and differences between gut, brain and dental pulp stem cells and their related CSCs, placing a particular emphasis on both their shared and distinctive cell markers, including the expression of pluripotency core factors. We discuss some of their similarities and differences with regard to oncogenic signaling, telomerase activity and their intrinsic propensity to degenerate to CSCs. We also explore the characteristics of the events and mutations leading to malignant transformation in each case. Importantly, healthy dental pulp stem cells (DPSCs) share a great deal of features with many of the so far reported CSC phenotypes found in malignant neoplasms. However, there exist literally no reports about the contribution of DPSCs to malignant tumors. This raises the question about the particularities of the dental pulp and what specific barriers to malignancy might be present in the case of this tissue. These notable differences warrant further research to decipher the singular properties of DPSCs that make them resistant to transformation, and to unravel new therapeutic targets to treat deadly tumors.

## 1. Introduction

Adult multipotent stem cells are responsible for renewing cell populations in the different organs of the body. The physiology and proliferative activity of these very different populations of organ-specific stem cells are adapted to fulfil the different requirements of the host tissue. Tissues differ markedly in their rate of mature cell turnover: there exist some tissues with high cell turnover activity owing to very high stem cell activity, whereas others have very low rates of cell turnover owing to a relative scarcity and/or quiescence of their adult stem cell populations.

The intestinal epithelium is the tissue with the highest adult cell renewal rate in mammals [1]. Millions of enterocytes are shed from the gut every day, which have to be replaced with new cells. The adult intestinal epithelium is constantly renewed by a population of cells located in the base of the Lieberkühn crypts: the ISCs, or intestinal stem cells. These are adult multipotent stem cells that respond very quickly to regenerative niche signals, and divide every 24 h to generate a new population of transit-amplifying cells, which gradually migrate towards the top of the villi whilst differentiating into different cell lineages [2]. The highly proliferative activity of ISCs allows for a practically complete replacement of the intestinal villi and crypt epithelial cells in a period of a few days. This turnover rate may be even accelerated in the case of presence of gut parasites, where it contributes to parasite expulsion [3].

On the diametrically opposite scenario, we find the central nervous system (CNS) with a very low self-renewal rate. Most cells in the brain and spinal cord are postmitotic neurons and glial cells. Indeed, the very existence of neurogenesis in the adult human brain was widely questioned by the neuroscientific community until very recently [4]. However, nowadays it is accepted that new neurons are generated throughout the whole life of the human brain, by activation of endogenous neural stem cells, or NSCs [5]. This natural neuron renewal process takes places mainly in a region of the limbic system called the hippocampus, a brain structure involved in memory consolidation [6]. Gliogenesis is known to be more widespread than neurogenesis, and it can take place in both the gray and white matter parenchyma, to ensure the renewal of postmitotic oligodendrocytes and astrocytes [7]. Both neurogenesis and gliogenesis are known to be relatively quiescent processes in the healthy mammal CNS, but they increase sharply after CNS injury [8,9]. However, there is controversy about whether the exacerbated activation of NSCs and the consequent reactive gliosis following brain injury are harmful or beneficial events for the restoration of normal CNS function [10].

Despite the great cellular and physiological differences between the healthy gut and CNS, both organs can give rise to very aggressive types of cancers. Colorectal cancer (CRC) is one of the most abundant cancers in the world and develops from the epithelial cells lining the colon or rectum of the gastrointestinal tract. As in other tumors, colon cancer cells are morphologically heterogeneous, differing in markers expression, proliferation capacity, tumorigenicity and chemotherapy resistance [11]. Glioblastoma, on the other hand, is a type of stage IV human brain cancer with the poorest prognosis owing to its very high ability to spread and infiltrate into brain parenchyma, thus hampering its total eradication by conventional therapies [12,13,14]. Both CRC and glioblastoma have in common the ability to relapse after surgical removal, which is attributed to the presence of cancer stem cells (CSCs) within the tumor.

The Cancer Stem Cell theory states that tumor growth is fueled by small numbers of CSCs hidden within the bulk of the tumor mass [15]. Much as normal cell renewal in healthy adult tissues depends on activation and proliferation of their endogenous stem cells, cell renewal in malignant tumors would depend on the activation of CSCs [16]. This theory explains clinical observations, such as the recurrence of tumors after initially successful therapy, and the phenomena of tumor dormancy and metastasis [17]. These CSCs or tumor-initiating cells were first described in teratocarcinomas that contained highly tumorigenic cells that, as single cells, could differentiate into multiple non-tumorigenic cell types [18]. Accordingly, the most accepted view of the theory of CSCs and tumorigenesis is that CSCs arise from and/or are closely related to normal adult multipotent stem cells, which undergo a process of transformation owing to the combined effect of gene mutations and cellular niche perturbation. Thus, understanding how adult tissue-specific stem cells behave in the healthy adult body, and how this homeostasis is lost under specific circumstances, is of paramount importance for the study of the different types of cancer.

The dental pulp shows a cellular renewal rate in between the gut epithelium and the brain parenchyma. Dental pulp tissue is of relevance to carcinogenesis because no cases of malignancies primarily originating in that location have ever been reported, at least since 1937 [19]. Back in the late nineteenth century it was common to diagnose putative dental pulp neoplasms as “pulpitis chronica sarcomatosa”, which was associated with bacterial infections and poor dental hygiene. However, closer examinations later on revealed that most, if not all, of those cases were not related to malignant neoplasms per se, but to a colonization of the exposed dental pulp space by the gingival epithelium [20].

It is unclear whether this dental pulp resistance to oncogenesis owes simply to the physical constraints of the reduced space of the dental pulp chamber, which would prevent a minimum tumor growth required for dissemination, or rather to other so far unknown factors [19]. The dental pulp contains its own stem cells: dental pulp stem cells (DPSCs), which have a particularly high self-renewal and multilineage differentiation capacity [21]. Interestingly, one surprising feature of DPSCs is that these cells are extremely resistant to anaerobiosis and lack of nutrients, as clearly demonstrated by some reported facts, such as their capability to survive and proliferate to confluency after travelling for more than one week in parafilm-sealed culture flasks under ambient temperature shipping conditions [22]. DPSCs in the healthy dental pulp are known to localize into neurovascular bundles containing nerves and blood vessels [23] and they are responsible for renewing populations of mature fibroblasts, odontoblasts, and myelinating Schwann cells [24]. DPSCs can generate complete dentin–pulp complexes in vitro and in vivo [21,25,26], and are also induced to activate and proliferate after dental injury [27].

Arguably, the tissues with the highest resemblance to the dental pulp would be embryonic mesenchymal tissues and also adult loose connective stromal tissues. Soft stromal connective tissues can be found distributed throughout the human body, and enriched in some particular locations including the skin, the bone marrow, the adipose tissue, among others [28,29]. These loose connective tissues can also give rise to soft tissue sarcomas with a very low incidence in the human population [30,31]. Notably, all of these tissues also home their own resident multipotent mesenchymal stem cells (MSCs), which share a great deal of characteristics, but also present important differences, with DPSCs. The systematic comparison between MSCs and DPSCs has already been the main topic of many excellent reviews [28,32]. It is compelling that all those connective tissues with so many resemblances at the structural and cellular level present so little rates of malignancy, which again suggests that there might well be other factors apart from mere physical isolation that would explain the absence of human cancers originating in the dental pulp.

In this review, we will discuss similarities and differences between three different stem cell types arising from different embryonic origins; ISCs and their corresponding colorectal cancer stem cells (CCSCs) from the endoderm, NSCs and their corresponding glioma stem cells (GSCs) from neuroectoderm, MSCs and their corresponding CSCs from mesoderm, and DPSCs from the neural crest. Their responsiveness to oncogenic signaling, expression of specific cell markers and cell pluripotency core factors, telomerase activities, and resistance to oncogenic transformation will be addressed. Interestingly, no dental pulp CSCs have ever been described in the scientific literature, which brings the attention to what specific characteristics of dental pulp cell biology might be responsible for the resistance of DPSCs to malignity. Could we attribute this to the differential expression of a particular cellular marker, or to any other physiological characteristic? Is there any CSC marker whose lack of expression in dental pulp cells could help explain why those cells are so little tumorigenic? Finally, is there any particular marker at all which can be regarded to be genuine of CSCs and excluded from all the rest of normal healthy stem cells? Those were the kind of questions that we set out to address in this review.

It should be noted that these mentioned stem cell types are able to survive and grow in vitro using the same type of culture media (Figure 1). So, they are not so different from each other, at least in terms of their minimum requirements of cell signaling and metabolism. Another important shared characteristic of these cells is that the expression of stemness markers in NSCs, MSCs, DPSCs and CSCs is normally promoted by their growth in serum-free media, whereas cell differentiation is normally induced when switching these cells to serum-containing media [33,34,35,36,37,38]. 

## 2. Cell Markers and Pluripotency Core Factors

### 2.1. Cell Surface or Membrane Markers

Searching for similarities between different types of normal and aberrant stem cells, numerous cell surface markers with different functions were found to be associated with the stemness characteristics of CCSCs (see Table 1). 

The first identified marker of stem cells and early progenitors in the mouse small intestine was CD133, a transmembrane glycoprotein, also known as Prominin-1 [39]. More recent studies found that CD133 was also expressed in a subpopulation of CCSCs [40,108,109] and these cells were present in metastasis and angiogenesis of colorectal adenocarcinoma [41,108]. Moreover, other reports using subcutaneous injection into immunodeficient mice, after comparison of CD133+ versus CD133- populations showed that CD133- ones were unable to form tumors or grow as undifferentiated tumorspheres [110]. Similarly, CD133 was one of the first reported markers of brain CSCs [45]. This marker was identified in brain stem cells of fetal [111] and premature infants [112] but initially reported as absent in adult human NSCs, contrary to murine NSCs [42,43,113]. Interestingly, CD133 has been postulated as an embryonic stem cell marker [44] and its presence or absence may suggest a different cancer origin accordingly to the cell transcriptional profiles. CD133+ brain CSCs behave similarly to fetal neural stem cells forming tumorspheres whereas CD133- brain CSCs display a semi adherent growth and its transcriptional profile is similar to adult NSCs [114].

CD133 expression has not yet been thoroughly evaluated for DPSCs. One study reporting a positive staining of DPSCs to CD133 relied exclusively on immunocytochemical characterization, which showed no clear membrane staining [46]. Other studies reported an absence of expression, as assessed by flow cytometry [47]. All these observations make CD133 an interesting potential marker for tumorigenic susceptibility across different tissues, and it would be interesting to address unequivocally whether CD133 is expressed or not in DPSC cultures, and whether this expression changes or not with different experimental conditions. Another comprehensive flow cytometry assessment also revealed the absence of CD133 in the more DPSC-related human bone marrow MSCs [115]. However, CD133 expression could be detected in CSC-like cells derived from malignant fibrosarcomas [116]. 

All this evidence combined points to CD133 as an interesting marker to identify CSCs in a large variety of tissues. However, some controversy remains with regard to the adoption of CD133 as a genuine universal CSC marker [117]. CD133 expression in CSCs has been related to particularly aggressive phenotypes but, as shown by different reports, this CD133 marker is also expressed by some healthy stem cell types such as ISCs and NSCs of the gut and the brain. However, it is precisely those organs, which give rise to very aggressive types of cancers, are also the ones that apparently show a higher basal CD133 expression in their resident stem cells, contrary to loose connective tissues, and the dental pulp in particular. This brings another issue: may eventually a positive relationship be established between the basal expression of CD133 in healthy stem cells and the malignity of neoplasms arising from CSCs in different tissues? This may deserve closer investigation. For instance, if CD133 expression was more consistently and comparatively assessed between different human stem cell types to fill in the existing information gaps, this could decisively help to clarify this question.

Another interesting surface marker of NSCs and neural CSCs is CD15, also known as SSEA-1 or Lewis X (Lex). This embryonic stem cell marker is expressed by some multipotent murine and human stem cells, and also by human CSCs [48,49,51,52,53,54,64,65,118]. Interestingly, CD133 and CD15 coexpression in a same cell (NSC or CSC) is very rare [53,119,120] with the exception of primitive neuroectodermal tumors which show higher clonogenicity for CD133+/CD15+ than CD133-/CD15+ cells [121]. CD133+ and CD15+ cells are sensitive to killing by oncolytic herpes simplex viruses [122]. In the intestine, CD15 expression has been found in Paneth cells in the vicinity of stem cell niches [123] but it is yet not clear whether CD15 is expressed or not by ISCs. CD15 expression is progressively increased during colon cancer development [48]. CD15/SSEA-1 has also been found to be expressed by both MSCs and DPSCs, as assessed by flow cytometry and RT-PCR. The expression of CD15/SSEA-1 was also found to be higher in DPSCs than in bone marrow MSCs [124]. Moreover, this embryonic marker was clearly upregulated in DPSCs after experimental conditions that increase the stemness of these cells, such as Wnt/ß-catenin activation [47,54]. The expression of CD15 has also been identified in fibrosarcoma-derived cells [125]. It would also be very interesting to compare the relative levels of CD15 expression in CSCs vs. MSCs and/or DPSCs to further validate the utility of CD15 as a CSC marker.

Another surface receptor that is present in human NSCs and CSCs is the Leucine-rich repeat-containing G-protein coupled receptor 5, or LGR5 [57,58]. LGR5, also known as GPR49, works as an important regulator of canonical Wnt/ß-catenin signaling and it physically interacts with Wnt Frizzled-5/LRP6 receptors [126]. LGR5 binds to R-spondins, which are secreted Wnt activator protein ligands. The LGR5 receptor has been postulated as a widespread CSC marker [127]. Interestingly, LGR5 is required for the tumorigenicity of glioblastoma cells [59]. LGR5 is also expressed by healthy colorectal ISCs [55] and its expression is maintained in CCSCs [56]. LGR5 is considered an ISC cell cycle marker and is related to survival, proliferation, and differentiation [128]. LGR5 can also mediate integrin signaling through MyoX and integrins present at LGR5 cytonemes [129,130]. Some subsets of MSCs have been shown to express LGR5, although the expression of this marker seems to be highly dependent on the mesenchymal cell source [131]. The expression of LGR5 in DPSCs also remains to be fully elucidated, although one report showed a strong expression of LRG5 in the dental pulp and the odontoblastic cell layer of permanent teeth [132]. Notably, the loss of expression of some specific LGR5 splicing variants has been associated with a poor prognosis in soft tissue sarcomas [133]. It is yet unclear how this differential expression of LRG5 isoforms may promote the development of human cancers, particularly with regard to the regulation of Wnt/ß-catenin activity.

The CD184 receptor, also known as Fusin or CXCR4, is a G-protein coupled receptor expressed by both brain NSCs and CSCs [79,80]. CXCR4 expression increases in vivo glioma perivascular invasion capacity [134]. In the gut, this receptor has been used as a marker to identify the CCSC population, together with LGR5 [135]. Furthermore, colon cancer cells double positive for CD133+ and CXCR4+ exhibit metastatic potential and their presence is linked to poor prognosis [78]. CXCR4 activation induces several cellular responses ranging from gene transcription and chemotaxis to cell survival and proliferation [136]. CXCR4 is amply expressed by human MSCs and DPSCs, where it contributes to stimulate their migration and chemotaxis, through activation of PI3K/AKT and Wnt/ß-catenin pathways [81,137]. In all cases, the activation of CXCR4 in stem and/or cancer stem cells is linked to the acquisition of a migratory cell phenotype and/or metastatic ability. 

The transmembrane glycoprotein, CD166 also known as ALCAM (activated leukocyte cell adhesion molecule) is an adhesion protein binding to the ECM and is expressed in both ISCs and CCSCs [60,61]. CD166 is also expressed by plastic-adherent human MSCs and DPSCs [64,65]. CD166 is not present in NSCs, nor in neural stem-like cells induced from conversion of bone marrow stromal cells (CD166+) [138]. The association between CD166 expression and poor prognosis of colorectal cancer is not yet sufficiently elucidated, with studies reporting different results [139]. However, in CD133+ GSCs, CD166 has been shown to increase cellular invasion [63]. The acquired expression of CD166 in GSCs could be associated with a higher migration and dissemination capacity, by the conversion to a more mesenchymal-like migratory phenotype. Under this viewpoint, an increased expression of CD166 in CCSCs could also be regarded as a contributing factor to explain the high tendency of CRC to form secondary metastases.

There are other mesenchymal stem cell markers associated with a higher tumor invasiveness. CD44 is a hyaluronic acid-binding surface receptor expressed in DPSCs, GSCs and CCSCs, but not in their corresponding healthy ISCs. Thus, CD44 constitutes a marker for prediction of hepatic metastases and poor prognosis in CRC [66,67,68,69,70]. CD90, also known as THY-1, is a GPI-anchored adhesion protein of the immunoglobulin superfamily, which is another widespread mesenchymal marker related to poor prognosis in many cancers [140]. As expected for a mesenchymal cell marker, CD90 is also expressed by DPSCs [71,75]. CD90 is a candidate marker for GSCs, but its expression is completely absent from healthy brain tissues [141]. Interestingly, CD90 has also been involved in the proliferation, migration and adhesion of human glioma-associated mesenchymal stem cells [74]. The cases of CD184, CD44, CD90 and CD166 constitute typical examples of mesenchymal surface proteins involved in cell chemotaxis and adhesion, whose expression is associated with the emergence of CSC phenotypes and particularly poor prognosis in many human cancers. However, because of the prominent expression of these markers in several types of healthy stem cells, such as MSCs and DPSCs, they also could not be considered to be genuine markers of CSCs.

### 2.2. Cytoplasmic Markers

One of the most prominent proteins required for self-renewal of NSCs is the intermediate filament Nestin [142]. Nestin is also expressed in several types of cancers [143], and also by CD133+ brain CSCs [85]. Nestin abundance is significantly correlated with prognosis, clinicopathological features and the histological grade of the glioma in patients [83,144]. Nestin+ tumor cells have been observed to be the origin of tumor regrowth after chemotherapeutic treatment with the alkylating agent temozolomide [145]. With respect to CRC, Nestin expression is upregulated in stromal cells and its knockdown inhibits migration and cell cycle arrest at S phase, thus halting cell proliferation [146]. GFP-tagged Nestin protein of several cancer cell lines including CRC revealed a high GFP expression in proliferating endothelial cells and nascent blood vessels in the growing tumors [147]. DPSCs in vitro are also practically 100% Nestin+ [75]. The widespread expression of neural markers by DPSCs is associated with the neural crest origin of these cells. In fact, contrary to other mesoderm-derived mesenchymal stem cells, such as those obtained by the bone marrow or the adipose tissue, DPSCs have been reported to show a much better capacity to differentiate to neuronal and glial cells [75,148,149] and these reasons partly account for the expectation raised by DPSCs as a non-conventional source of stem cells for neural regeneration [150,151]. Another important feature of DPSCs related to their use for neural cell therapy is their ability to secrete neurotrophic factors [75,152] and differentiate to endoteliocytes and pericytes to generate new blood vessels within the CNS [100].

The intracellular RNA binding-protein MUSASHI is a marker of poor prognosis in many human cancers and regarded as a putative marker for CSCs. It is known to be expressed by ISCs [86], and it is overexpressed in CRC, where its levels correlate with other stem cell marker of the intestinal epithelium such as ß1-integrin and LRG5, suggesting its involvement in CCSC generation [77,153]. MUSASHI is involved in the maintenance of adult stem cell fate, and also expressed by NSCs and GSCs [40,154,155] where it participates in enhancing tumoral cell migration [156]. MUSASHI has also been shown to be expressed by DPSCs and other stem cells of the oral cavity, and its expression was reported to increase in response to osteogenic differentiation [90]. There exist yet no reports of MUSASHI expression in soft tissue sarcomas.

One marker whose loss is related to poor prognosis in human cancers is the Phosphatase and Tensin Homolog (PTEN). This enzyme is critical for stem cell maintenance and PTEN deficiencies can cause the development of CSCs [157]. It has been observed that PTEN loss reprograms healthy stem cells to adopt a glioblastoma stem cell-like phenotype [106]. PTEN is also involved in the migration of precursor cells [158] and it is expressed in adult NSCs and progenitors [102,103]. Its deletion or loss of function has been reported to alter neurogenesis and provokes cellular alterations in adult hippocampal neural progenitor and stem cells [159]. PTEN has also been involved in the control of the proliferation rate and number of ISCs, and similar to what is observed in other regions, the absence or dysfunction of PTEN provokes an intestinal polyposis due to an excessive cellular proliferation becoming a precancerous neoplasia [104].

Recent investigations show that DPSCs present unusually high levels of PTEN expression [107]. This accounts for critical differences between DPSCs and other related multipotent stem cells, such as mesodermal MSCs. In a comparative study between DPSCs and bone marrow MSCs, the high levels of PTEN expression naturally present in DPSCs were shown to downregulate the oncogenic PI3K/AKT pathway, thus contributing to an increased osteo/odontogenic capacity, at the expense of a diminished tumorigenic capacity. Remarkably, when both bone marrow MSCs and DPSCs were transfected in parallel with pRB and CMYC oncogenes, MSCs readily acquired a tumor phenotype whereas DPSCs could only be induced to transform when PTEN was also simultaneously inhibited [107]. Due to its role as a tumor suppressor gene, it might be postulated that at least some of the resistance of DPSCs to oncogenesis may be attributed to a high expression of PTEN. Another implication of this hypothesis would be that stem cells from other body locations could also be more vulnerable to transformation because of an insufficient PTEN expression.

### 2.3. Nuclear Proteins

One of the principal characteristics of CSCs that distinguish them from the rest of tumor cells is their overexpression of nuclear transcription factors traditionally associated to stemness and pluripotency. SOX2 is one of the principal core factors related to cell pluripotency [92] and encodes a transcription factor member of the SRY-related HMG-box (SOX) family. In CRC, SOX2 positive cells were found to display several characteristics of CSCs, together with a decreased expression of the intestinal epithelial marker CDX2, contributing to a poor prognosis [91]. OCT4A is another core factor which has been linked to chemoresistance of colon CSCs [97] and NANOG has been recently related to colony formation and growth of CRC cells [101]. Notably, SOX2 is expressed by both healthy NSCs and GSCs [52]. Interestingly neural progenitor cells have also been reported to express mRNA for NANOG and OCT4 [98]. It should be taken into account that the expression of these stemness factors, together with others such as KLF4, leads to the development of induced pluripotent stem cells (iPSCs) with tumorigenic capacity [99] and SOX2, OCT4 and NANOG are all pluripotency markers [93] that are found in circulating tumor cells present in the blood of patients with glioblastoma [94].

Nevertheless, it should be noted that the mere coexpression of these three markers per se does not necessarily induce CSC-related cell phenotypes. Healthy non-tumorigenic DPSC cultures, for instance, also show coexpression of SOX2, OCT4A and NANOG [54,160]. Moreover, the expression of these pluripotency core factors rises in DPSCs subjected to activation of Wnt/ßcatenin signaling to enhance their stemness potential, but without leading to cell transformation [54,95,96]. Wnt/ßcatenin signaling was also shown to promote the maintenance of pluripotency in embryonic stem cells [118].

## 3. Oncogenic Signaling

### 3.1. Wingless (Wg)-Related Integration Site (Wnt) 

Wnt genes are extensively conserved between invertebrates and mammals, thus highlighting the importance of this signaling pathway to regulate cell development and gene expression [161]. Once secreted, Wnt proteins bind to specific membrane Frizzled/LRP5-6 coreceptors on the target cell. These events lead to the membrane recruitment of an intracellular multiprotein complex containing (among others) AXIN2, APC and Glycogen Synthase Kinase-3ß, which causes the inactivation of the latter enzyme, and the dephosphorylation and eventual accumulation of ß-catenin protein, a fundamental transcriptional coactivator of Wnt target genes. Wnt/ß-catenin signaling can be potentiated by simultaneous activation of LRG5 by R-spondins [126], or alternatively weakened by other LRP5-6 ligands such as DKK-1 [162]. Many malignant cancer cells show a pathological hyperactivation of canonical Wnt/ß-catenin signaling [163] and mutations that promote a constitutive activation of the Wnt pathway, such as inactivation of APC or DKK-1, very often lead to colorectal cancer [164,165]. ISCs are positive for LGR5 [55] and AXIN2 [166], which underscores the importance of canonical Wnt signaling in controlling the homeostasis of these cells [55,166,167]. On the contrary, the loss of Wnt function is associated with defects in epithelial cell renewal in many organs, including the gut [168,169]. 

Wnt activity in NSCs regulates their homeostasis and adult hippocampal neurogenesis [170]. In human glioma cells, it has been described as an important regulator of cell proliferation [171,172,173]. Thus, both healthy ISCs and NSCs are sensitive to Wnt signaling, and an excessive Wnt/ß-catenin hyperactivation is believed to promote the transformation of stem cells to CSCs in both cases [164]. This parallelism shows a common ground between ISCs, NSCs, CCSCs and GSCs [169,174,175,176]. Strikingly, the “need” for the Wnt pathway and the struggle this creates between cancer cells and neural cells reaches to levels of “vampirization” in which neurons end dying from the subsequent enwrapping and Wnt receptor depletion taken by the squeezing and invading glioma cells. Using this strategy, cancer cells manage to increase their available space and their own proliferation and infiltration capabilities within the brain [177]. Furthermore, Wnt activity is required for self-renewal of GSCs [178]. Taking everything into account, it is not surprising that the targeting of the Wnt pathway has been recently regarded as a high priority for therapeutic advances [163].

Another cell type with a high sensitivity to Wnt signaling is the DPSC. It was recently shown that even very short-term applications of Wnt-3a in DPSCs are associated with an increased self-renewal and an enhancement of their stemness properties [54]. Moreover, the increase in multilineage differentiation potential in DPSCs is associated with a deep remodeling of DPSC physiology at both the metabolic and epigenetic level [95,96]. However, Wnt/ß-catenin activation caused only a modest increase in the self-renewal capacity of DPSCs [96]. It is currently unknown whether the non-tumorigenic phenotype of DPSCs could have any relationship with a tighter regulation of Wnt signaling, in comparison with CCSCs or glioma CSCs [174]. It should also be taken into account that not only the tumor cells themselves, but also stromal cells around the tumor may also secrete additional factors activating the Wnt/ß-catenin signaling pathway, to promote tumor cell invasion and metastasis. The contribution of stromal cells, especially fibroblasts, endothelial cells and pericytes, appears to be very relevant for the progression of both malignant CRC and glioblastoma [179,180].

### 3.2. Transforming Growth Factor Beta (TGF-ß) Signaling

TGF-β superfamily signaling plays key roles in cell differentiation and proliferation [181], and comprises over 30 different members including activins, nodals, bone morphogenetic proteins (BMPs), and growth and differentiation factors (GDFs) (see review [182]). TGF-β pathway activity is able to lengthen the progression of the cell cycle in aged NSCs [183]. This signaling also increases in the neurogenic niches during aging or after a high dose of radiation inducing the quiescence of NSCs [184]. However, it may also induce epithelial–mesenchymal transition (EMT) of normal cells to acquire migratory and stem cell properties [185]. TGF-ß1 protein is known to be up-regulated during ageing, brain lesions or during neurodegeneration [186] but is also involved in the development and progression of high-grade gliomas [187,188]. TGF-β also promotes tissue invasion, angiogenesis and evasion from immune attack [189,190].

TGF-β signaling also regulates stemness of normal stem cells and CSCs [191]. Indeed, during development Activin and Nodal proteins regulate NANOG expression maintaining cellular pluripotency in human and mouse embryonic stem cells [192]. The importance of TGF-ß signaling in the maintenance of stemness of DPSCs needs further clarification. One report showed significant expression changes of expression in several TGF-ß-related genes after induction of DPSC differentiation to osteo/odontoblasts by standard pharmacological protocols. Specifically, Activin A, TGF-ß1 and TGF-ß2 expression were shown to be downregulated, but TFG-ß receptors II and III upregulated, after DPSC osteoblastic differentiation [193].

In the large intestine, it is assumed that the TGFβ/BMP signal gradually increases along the intestinal axis of the villi of the crypt, while the gradient towards the base of the crypt decreases, thus inhibiting the regeneration of stem cells and supporting the differentiation of epithelial cells, thereby playing a vital role in balancing the effect of Wnt signaling on intestinal homeostasis [194]. However, dysregulation of TGF-β signaling is involved in cell proliferation, differentiation, migration and apoptosis, and it could lead to the development of CCSCs [195]. Interestingly, a cross-talk between TGF-β signaling and the R-spondin/LGR5 axis was reported in CRC cells, where LGR5-induced TGF-ß activity in tumor cells was associated with a decreased tumor invasion and metastasis [196]. However, the activity of the TGF-β pathway in stromal cells is associated with a higher risk of metastasis in CRC, and pharmacological inhibition of TGF-ß receptor I impairs tumor metastasis in CRC [197]. Once again, these results show the involvement of TGF-ß signaling to promote an oncogenic tissue microenvironment and highlight the importance of the crosstalk between tumor and stromal cells to sustain cancer malignancy.

## 4. Telomerase Activity

The regulation of cellular telomerase activity depends on the transcriptional control of its two essential components, hTERC (RNA component) and hTERT (reverse transcriptase component) [198]. Wnt/β-catenin signaling was shown to have a positive regulatory effect on the expression of telomerase reverse transcriptase (hTERT) and CSC-related proteins [199]. In turn, telomerase directly modulates Wnt/β-catenin signaling, by activating quiescent stem cells [200]. Importantly, the emergence of CSCs is promoted by the overexpression of hTERT [201,202,203]. Interestingly, it has been reported that a mutated TERT fragment is able to induce brain cancer stemness independently of its telomerase activity [204]. Furthermore, telomere dysfunction promotes tumorigenesis by inducing chromosomal instability in tumor initiating cells (see review [205]). Chromosomal instability is a source of genetic variation, favoring tumor adaptations to stressful environments and cytotoxic anticancer drugs, contributing to the progression at multiple stages of tumor evolution [206,207]. Normally, telomerase activity is downregulated after human brain embryonic development even in adult multipotent stem cells [208,209]. However, multipotent stem cells such as ISCs, MSCs, DPSCs and NSCs all present a basal telomerase activity and hTERT expression [210,211,212,213].

Brain telomerase activity in adult mice has been found to be restricted to the subventricular zone and olfactory bulb [214]. It plays an important role in cell proliferation in the adult but not in embryonic NSCs [215]. Telomere length has also been demonstrated to be important for neuronal differentiation and neuritogenesis [216] (see also review [217]). Its deficiency leads to a compromised olfactory bulb neurogenesis [215] although NSCs lose telomerase activity upon differentiation into astrocytes [218]. DPSCs also lose progressively their telomerase activity upon their spontaneous in vitro differentiation to osteoblastic/odontoblastic cells in conditions of high culture passages [212].

It should be emphasized that telomerase is reactivated in some malignancies such as CRC and most of brain cancers [219]. However, the mere absence of telomerase activity does not guarantee cellular resistance to oncogenic transformation, because CSCs may also use a mechanism of alternative lengthening of telomeres (ALT) [220,221]. ALT preserves telomeres by homologous recombination machinery independently of hTERT and hTERC [222]. ALT has been reported to be present in 10–15% of human cancers, including GSCs [52,223], but also in colon cancer cells with BRCA2 deletion [224] or hereditary and sporadic colon cancer [225]. ALT has not been reported to date in NSCs nor ISCs, suggesting that the origin of ALT in CSCs of brain and colon cancer could also be related to a dedifferentiation process from somatic cells [225]. ALT has not been yet reported for DPSCs. However, this mechanism is very active in malignant tumors of mesenchymal origin [226,227]. It remains to be studied whether the ability to activate ALT could constitute another important difference between DPSCs and MSCs.

## 5. Pathways and Obstacles to Malignant Transformation: The Surprising Case of DPSCs 

By making a systematic comparison between normal and cancer stem cells of different embryonic origins, we have identified a set of differential markers that are all present in brain and gut CSCs but absent or at least not yet well defined in other healthy stem cells (Figure 2). Interestingly, the first conclusion of this analysis is that a surprisingly high number of CSC markers are also expressed by some normal stem cells. DPSCs deserve an extra mention here, because up till now there exists no CSC marker at all that has been conclusively characterized to be absent from these cells. This brings the response to the question of the title of this manuscript, whether there was such a thing as a genuine CSC marker. According to the available information, we can conclude that as yet there exists no such marker, because practically all the markers mentioned in this review, that were so far regarded as more or less specific of CSCs, are also known to be expressed by DPSCs, with the only possible exceptions of CD133 and (less likely) LGR5.

Normal and tumor stem cells share some important features. Due to this high resemblance and according to the CSC theory, normal healthy stem cells in the body would be the most sensitive cellular targets to undergo malignant transformation, although this does not rule out a possible contribution from other more differentiated types of cells. CSCs are a small subpopulation of undifferentiated tumorigenic cells inside the tumors with malignant phenotypic characteristics [230]. In the brain, NSCs and early progenitor cells have been reported to give origin to glioblastoma [231]. ISCs can acquire mutations in Wnt-pathway-activating genes that may initiate cancer [232]. Furthermore, CCSCs and brain CSCs acquire mutations in oncogenes and tumor suppressor genes such as PTEN, TP53 and RB1, which confer them different abilities including: stemness, producing actively proliferating cancer progenitor cells in their niches, multidrug and apoptosis resistance and enhanced DNA repair capacity [196,233]. Other aspects that have been proposed to lead to cellular transformation are the alteration of several signaling pathways such as EGFR and INK4a/ARF [234] or the stromal cell recruitment and production of proinflammatory signals in the niche [235], which is a very old concept that has been retaken over the last decades [236,237].

One of the most intriguing questions about the dental pulp is that, contrary to many others, this tissue is extremely unlikely to generate malignant neoplasms. The case that the dental pulp niche would just be more protected against mutagens and inflammatory reactions does not seem to hold very well to explain this phenomenon. For instance, dental pulpitis stands arguably as quite common and one of the most exacerbated inflammatory reactions in the whole human body [238]. Yet for all this evidence, there are no case reports of malignant tumors with an alleged origin in the dental pulp, at least since 1937 [239]. DPSCs seem to be placed in a particularly disadvantageous position to face all the looming threats that could contribute to transform a healthy stem cell to a CSCs. It could be somehow expected that the remarkable stemness of DPSCs would also contribute to boosting their transformation to particularly malignant cell phenotypes. DPSCs consistently express pluripotency core factors (SOX2, OCT4, NANOG, CD15, among others); they express adhesion proteins associated with metastasizing ability (CD44, CD90, CD166); they are highly responsive to signaling pathways (Wnt, TFG-ß) linked to oncogenesis; they are resistant to radiation; they resist anaerobiosis; they present hTERT activity, similar to many CSCs and contrary to many other healthy adult multipotent stem cells in the body. However, despite all these characteristics, DPSCs show a very low tendency to transform, not even after a forced overexpression of hTERT [240,241] and of oncogenic E7 protein from human papillomavirus [242]. 

If DPSCs share so many characteristics in terms of mitogenic potential, responsiveness to signaling pathways, and marker expression with CSCs, the next obvious question is: what makes them so resistant to neoplastic transformation? It has been traditionally argued that the reduced dental pulp space acts as a natural barrier preventing tumors from reaching a critical size for dissemination. In addition, the reactive differentiation of DPSCs to dentin-producing odontoblasts upon contact with tumor cells would trigger the sclerosis and calcification of the dental pulp chamber, thus minimizing even further the available space for tumor growth [19]. However, there are alternative explanations, such as a high expression of tumor-suppressor genes by DPSCs. This is clearly an aspect of DPSC biology that has not been yet sufficiently studied. May it be that DPSCs are equipped with a better genetic armor to avoid transformation? If we identify the genetic signatures that make DPSCs so resistant to oncogenesis, could we try to boost those characteristics in other more vulnerable and/or pre-neoplastic cells? We have previously shown the example of PTEN [107]. This accounts for critical differences between DPSCs and MSCs, with regard to oncogenic susceptibility. Among tumor suppressors expressed by DPSCs, we also include some microRNAs, such as let-7c, which are involved in the regulation of cell growth and differentiation via IGF/MAPK pathways [243]. Another particularly important aspect of DPSCs is their capacity to trigger very strong DNA damage responses to genotoxic stress [244]. In a comparative assessment of DPSCs and human dermal fibroblasts exposed in parallel to a genotoxic/cytostatic cisplatin treatment, DPSCs were found to activate the p53/p21 pathway more potently, thus leading to cell cycle arrest and a rapid onset of senescence or apoptosis [245]. This ability to readily shut down cell proliferation in response to DNA damage constitutes a fundamental defense mechanism against tumorigenesis, by avoiding the expansion of cells which have a compromised genome integrity [246].

Experimental evidence, such as the one mentioned before, demonstrates that it is very likely that there also exist genetic and/or physiological reasons for the inexistence of human dental pulp cancer and dental pulp CSCs. If the cancer stem cell theory holds true, then there would be a lot of valuable lessons to be learnt from these stem cells that naturally resist to transformation. Thus, the oncology research field might well find new inspirations by looking through the window of the dental pulp.

## 6. Conclusions

In this review, we have compared stem cells and cancer stem cells from different embryonic origins. Some multipotent stem cells in the adult body are believed to transform and generate CSCs which fuel very aggressive tumors with very poor prognosis, whereas other adult stem cells have so far never been involved in the generation of human cancers. The case of DPSCs is particularly intriguing because these cells express a large number of markers that have been identified and reported as highly characteristic of CSCs. In fact, none of those markers have yet been conclusively demonstrated to be absent from DPSCs. It is unclear where DPSC resistance to oncogenesis may come from: whether the physical constraints imposed by the dental pulp space, the high level of expression of certain tumor-suppressor genes, the capacity to discard defective cell progenies by inducing potent DNA damage responses leading to senescence/apoptosis, or a combination of all of these factors together. These aspects warrant further research with a view to gain new knowledge on the comprehension of stem and cancer stem cell biology. Furthermore, unraveling the mechanisms of DPSC resistance to oncogenesis might also open new therapeutic avenues and strategies to avoid or mitigate the malignant transformation of some deadly tumors.

## Figures and Tables

**Figure 1 biology-09-00426-f001:**
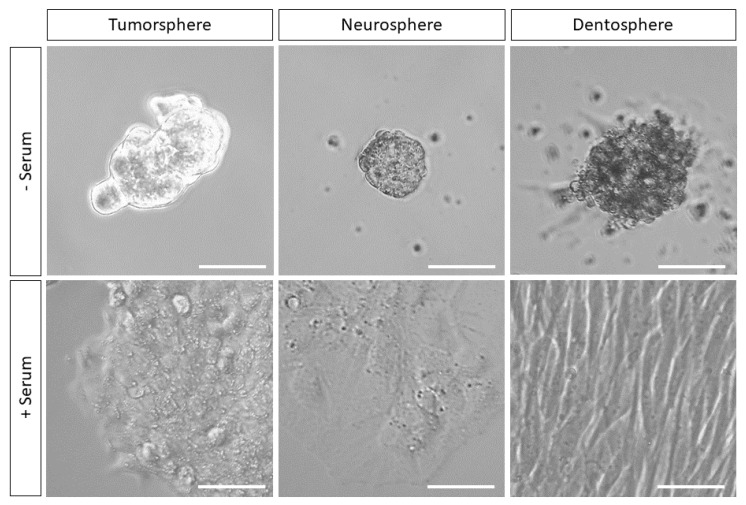
Stem cells and cancer stem cells (CSCs) from three different embryonic origins (endoderm, neuroectoderm, neural crest) can be maintained in the same cell culture media for long periods. Oncogenic CRC-derived SW620 colon cancer stem cells (CCSCs, left), and normal healthy neural stem cells (NSCs) (middle) and dental pulp stem cells (DPSCs) (right) can be grown under the same culture conditions in either the presence or absence of fetal serum. When grown on DMEM/F12 serum-free medium supplemented with basic fibroblast growth factor (bFGF) and epidermal growth factor (EGF), CCSCs, NSCs and DPSCs generate free-floating spheres (upper row) that can be in vitro maintained for several months. Cells from the same batches were grown in parallel with the same medium supplemented with 10% fetal bovine serum (bottom row), and in all these cases they generated plastic-adherent cell monolayers. Scale bars 50 μm.

**Figure 2 biology-09-00426-f002:**
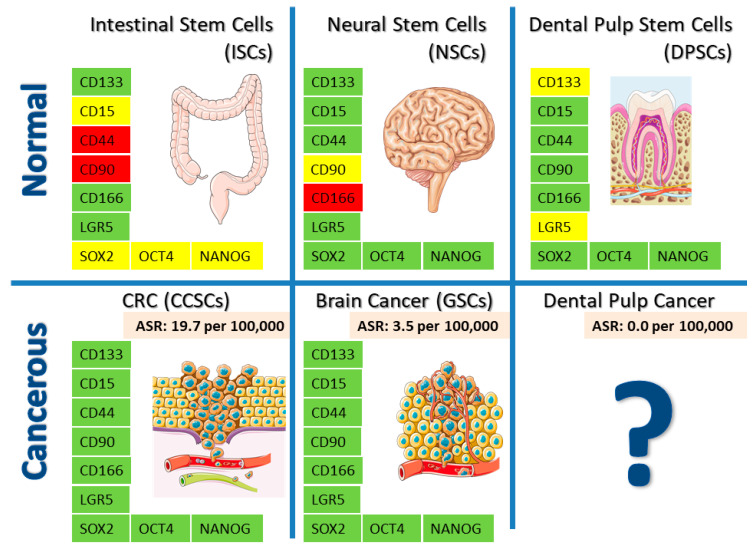
Comparison between ISCs, NSCs and DPSCs and their corresponding CSC derivatives. For the sake of simplicity, only markers with a differential expression are shown. Many of the markers expressed by gut and brain CSCs are also shared by normal healthy DPSCs, with the cases of CD133 and LGR5 as the only yet unknown exceptions. Healthy NSCs and DPSCs share at least 7 differential markers each with CSCs. ISCs show far fewer coincidences with CSCs in terms of marker presence/absence. Despite the great similarity in marker expression between different CSCs and DPSCs, there exist absolutely no reports showing phenotypes of CSCs originating in the dental pulp, and dental pulp cancer incidence worldwide is 0%. Markers in green color: confirmed expression. Markers in red color: confirmed absence of expression. Markers in yellow color: Not well defined or unknown expression. The yearly worldwide incidence of each cancer type for 2018 is expressed as Age Standardized Rates (ASR) per 100,000 people (pink color), as reported in [228] and [229]. Images credit: Creative Commons License Servier Medical Art by Servier (https://smart.servier.com/). All central nervous system cancers are considered within brain cancer rate. CRC: colorectal cancer. CCSCs: colorectal cancer stem cells. GSC: glioblastoma stem cells.

**Table 1 biology-09-00426-t001:** Expression of cell markers and core factors involved in cancer in ISCs, NSCs and their corresponding CSCs, and DPSCs. Green color represents confirmed expression as reported by the literature. Red color represents absent expression as reported by the literature. Yellow color implies that its expression is not well defined or needs to be further studied. “+” symbol implies the existence of expression-confirming reports, “-” symbol implies the existence of no-expression supporting reports and question mark “?” implies that the expression of that marker has not been yet thoroughly assessed for a particular cell type. “↑” or “↓” symbol refers to an upregulation/increase or downregulation/reduction in its activity/expression with respect to their normal cell counterparts. ISC: intestinal stem cells. CCSC: colorectal cancer stem cells. NSC: neural stem cells. GSC: glioblastoma stem cells. DPSC: dental pulp stem cells. CSC: cancer stem cells.

	Intestinal		Neural		Dental
	ISC	CSC/CCSC	NSC	CSC/GSC	DPSC
CD133	+ [39]	+ ↑ [40,41]	+ [42,43,44]	+ ↑ [43,45]	+/- [46,47]
CD15	?	+ [48]	+ [49,50]	+ [51,52,53]	+ [54]
LGR5	+ [55]	+ [56]	+ [57,58]	+ [59]	?
CD166	+ [60,61]	+ [61]	- [62]	+ ↑ [63]	+ [64,65]
CD44	- [60]	+ ↑ [66]	+ [67]	+ ↑ [68,69,70]	+ [71]
CD90	- [72]	+ ↑ [73]	- [62]	+ ↑ [74]	+ [75,76]
CXCR4	+ [77]	+ ↑ [78]	+ [79]	+ ↑ [80]	+ [81]
NESTIN	+ [82]	+ ↑ [83]	+ [84]	+ ↑ [83,85]	+ [54,75]
MUSASHI	+ [86]	+ ↑ [87]	+ [88]	+ ↑ [89]	+ [90]
SOX2	?	+ ↑ [91]	+ [52,92]	+ ↑ [52,93,94]	+ [54,75,95,96]
OCT4	?	+ ↑ [97]	+ [98]	+ ↑ [93,94,99]	+ [54,95,96,100]
NANOG	?	+ ↑ [101]	+ [98,102,103]	+ ↑ [93,94]	+ [54,95,96,100]
PTEN	+ [104]	+ ↓ [105]	+ [106]	+ ↓ [106]	+ [107]

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
