# Peer review of "Is There Such a Thing as a Genuine Cancer Stem Cell Marker? Perspectives from the Gut, the Brain and the Dental Pulp"

_biology, 2020, doi:10.3390/biology9120426_

Round 1

Reviewer 1 Report

The authors carefully addressed all my comments and thus further improved the quality of the review.

Author Response

Thank you very much for your feedback.

Reviewer 2 Report

This is a well-written, detailed and original review which summarizes and discuss relevant literature on differences and similarities among gut, brain and dental pulp stem cells and their related cancer stem cells, highlighting the fact that, so far, dental pulp stem cells have not been related to malignant tumors. 

Author Response

Thank you very much for your detailed feedback.

Reviewer 3 Report

The revised manuscript has been improved. However, my previous and only comment to the authors was about checking carefully all the references. I mentioned ref#100 as an example of an inappropriate reference. This reference (now#110) has not been modified. The authors claimed that they have performed a new careful revision of the reference field. Please explain why this reference is indeed relevant concerning Nestin ? (lines 264-265)

Author Response

We apologise for the error. We are totally agree with the reviewer, the reference #110 of Sunyoung Park in incorrect. Indeed there are two Park from the same year 2010. The correct reference is from Donghyun Park:

Reference 110 has been replaced with the correct one:

Donghyun Park et al. Nestin is required for the proper self-renewal of neural stem cells. Stem Cells. 2010 Dec;28(12):2162-71. doi: 10.1002/stem.541.

This manuscript is a resubmission of an earlier submission. The following is a list of the peer review reports and author responses from that submission.

Round 1

Reviewer 1 Report

Crende et al. present here an interesting review that has an engaging premise: something important about cancer could be learned by considering dental-pulp stem cells, which do not appear to give rise to malignancy, in contrast to stem cells from gut and brain tissue. This review examines these three types of stem cells (brain, colon, and dental pulp) and each of their propensity to transform into cancerous cells. The authors list many differentially expressed factors and markers. The authors highlight the intriguing point that dental pulp cells have little likelihood, if any, to develop into cancer cells, though they have many similar markers and highly cancer-prone stem cells. 

Overall, Crende et al. provide a review with a very long list (247) of references that mostly just surveys the literature, but thankfully the authors also infuse into the review a particular point of view that makes it more interesting to read for the non-CSC biomedical/cancer researcher. These major issues are listed below, and range from rethinking and thoroughly revising the visuals to further extrapolating upon, and wrapping up, the authors’ main thesis. 

Major concerns:

  1. The authors’ otherwise compelling thesis — that something exciting could be learned from comparing cancer-innocuous dental-pulp stem cells to brain/gut — is not really answered/sufficiently wrapped up by the end of the review. The authors’ thesis piques the reader’s interest, but then the reader is “left hanging” since it is not fully addressed in the review.
    1. It ultimately remains unclear what the focus/point of this review was. Perhaps it is to set-up the idea that DPSCs need further research because unlike other stem cells which give rise to very malignant cancer types, DPSCs don’t? Or to compare various types of naturally occuring stem cells and the cancers they can give rise to?
    2. The Title doesn’t actually reflect the review. It really must be revised to reflect the review as-written more accurately.
    3. Dental pulp cells are referred to as having “intrinsic characteristics” (line 94) or “other critical aspects” (line 511) or a “tight regulation of physiology” (line 535) that make them resistant to malignant transformation.  However, it is unclear what specific factors, proteins, etc., the authors are precisely referring to, aside from the example of PTEN offered at the end. These instances should be reworded or removed in order to accurately portray potential evidence-based reasons that DPSCs may be resistant to becoming cancerous.
  2. Although it is clear to me why dental pulp stem cells are noteworthy, since they do not seem to give rise to cancers, the authors should validate further why considering specifically brain and gut stem cells are the most logical contrasting choice for comparison. Without further validation of these other tissues’ stem cells, the choice of these two alternatives that do give rise to aggressive cancers seems to simply be a convenient most-severe choice of the authors. Are there tissues that are in many ways very similar to dental pulp in many ways (more similar than gut/brain), but that actually do give rise to cancers (even if not that severe) that would be informative to compare to DPSCs? Perhaps this tissue type would have just a small number of variables that makes the difference in terms of markers, etc.
  3. Table 1 is extremely confusing. It needs to be completely overhauled to be intelligible. 
    1. It is hard to know why there is information without citations in some locations. If it is not known, then the “?” symbol is used; if it is known that there is negative expression, that should also be cited, but often it is not. The presence of data but lack of a citation in many places is vexing.
    2. It is hard to tell the “+/–/?” signs from the citations; could the authors use color? How about reformatting the Table to be like a heat map, with red vs. green rectangles for each cell of the Table instead of + and – signs? 
    3. Also, the use of “negative/positive expression” is not clear; do the authors mean differentially expressed (up vs. down relative to normal cells or is it detectable vs. undetectable, or…)? If it is relative to normal/wild-type, then how is their normal-tissue example a relevant column if the cancer cells are based on expression level from this column? 
    4. It would also be helpful to add lines between tissue types to easily differentiate the boundary between them. 
  4. Figure 2 just seems to be a repeat of Table 1.
  5. The layout of Figure 2 is overly complex and hard to quickly understand. 
    1. For example, if the top row is normal, non-cancerous cells, why not have the word “Normal” or “Wild-type” or something as a label to the side of the chart, and “Cancerous” for the lower row? 
    2. There is too much redundancy of text; in particular, the cellular marker genes and boldface type are very confusing and it is way too hard to quickly perceive the authors’ points (in fact, I am not sure the points are clear even after scrutinizing the figure). I am sure the authors can apply more critical thinking to this chart and come up with a more streamlined presentation. One simple clarifying suggestion that reduces text and improves clarity would be to, for example, have “Nestin” stated just once on the left of the table and then use a + vs – across rows for each cell type to show if it is present/absent (and the authors may want to use ++ or something if it is thought to be very high). 
    3. If boldface indicates differing presencence/absence between tissues, why, e.g., is MUSASHI bold between ISCs and CCSCs yet in both cases it says “MUSASHI+” (in bold)??
  6. In Figure 1, the contents of and differences between the images is not as clear as possible. It would be helpful to add +serum and -serum or something similar on the side, and to add the cell type in addition to the existing labels. It may seem more logical to reverse the order such that the “wild type,” healthy cells are at the top. Finally, scale bars are absent from three of the images.

Minor concerns: 

  1. The Figure 2 legend has 2 redundant title-like statements. Also, the images used look like they are taken from a textbook or something? Should there be citations in this case; who owns/copyrighted (?) these images or did the authors actually draw them?
  2. The image quality of Figure 2 is slightly pixelated and would appear cleaner. The clarity of the figure would also be improved if the titles of the cell type were larger than titles of markers.
  3. Line 51: How are the presence of gut parasites and/or an increased turnover rate specifically acting as a natural defense mechanism?
  4. Line 90-91: before 1937 how many cases were reported?
  5. Line 234-240: seems out of place, should be new paragraph, new section, or repositioned into the conclusion when discussing DPSCs
  6. Line 248: how did the expression change? Does it increase or decrease? By how much?
  7. Line 249: why is DCLK1 interesting? Little is known about it, its role is "controverted" and nothing is known related to DPSCs
  8. Line 377/citation 186: cite more primary literature here
  9. Line 385-386 this is misleading because it implies that the absence of telomerase results in cellular resistance to oncogenic transformation
  10. Line 393: What is meant by “consistent”? Consistently high? low? average? compared to what?
  11. Line 437-438 therapeutic potential seems out of place, not relating to DPSCs
  12. Line 442 why is that logical?
  13. Line 474 what is meant by “today”? Reference from 2001?
  14. Line 488 what is “solid’ telomerase activity? Same level as something else? Increased relative to something else?
  15. Line 501: add PTEN to Table 1/Figure 2

Spelling and grammar concerns

  1. Line 16:  The conversion of the healthy stem cells
  2. Line 19:  However, this is not the case of the dental pulp
  3. Line 21: regarding to: remove “to”
  4. Line 25:  there exist almost literally no reports about
  5. Line 26: reword “This raises the question of what is so special about the dental pulp” to…
  6. Line 48: reword intense
  7. Line 49: a very few days: remove “very”
  8. Line 58: takes places: change to “takes place”
  9. Line 73: Glioblastoma: should not be capitalized
  10. Line 81: this CSCs: should be these CSCs
  11. Line 126: incomplete sentence
  12. Line 149: is “positivity” is the best way to describe the purported presence of CD133?
  13. Line 259: use of “differentiates” (meaning “is different from”) is confusing because it is in the context of stem cells, which can differentiate in the sense of specifying cell types.
  14. Line 388: to: change to “of”
  15. Line 409: reposition “from the latter” to immediately follow “gain
  16. Line 412 noteworthy: add “it is (noteworthy) that…”
  17. There are various sentences/sections which need to be reworked because they are repetitive.
  18. Proofreading is also needed to address some grammar/wording choices
  19. In general, more transition sentences are needed to better connect the flow of the paper.

Author Response

We are grateful for your overall positive opinion and appreciate your very valuable feedback to improve this manuscript. We have done our best to implement all your suggestions in this new version. 

Major concerns:

The authors’ otherwise compelling thesis — that something exciting could be learned from comparing cancer-innocuous dental-pulp stem cells to brain/gut — is not really answered/sufficiently wrapped up by the end of the review. The authors’ thesis piques the reader’s interest, but then the reader is “left hanging” since it is not fully addressed in the review.

We completely agree that the last manuscript version ended up leaving many open issues, particularly to what concerned the reasons of oncogenic resistance of DPSCs. It was our intention to highlight the fact that there is something in the dental pulp that protects its cells against transformation. However, the information is limited and this is clearly an area which needs further investigation.

Nevertheless, beyond piquing the reader’s interest we agree that we still needed to give more specific evidence. We have managed to complete this last section by adding additional information about the expression of tumor suppressor genes, and the onset of p53-mediated DNA-damage responses by DPSCs. The possible mechanisms responsible for oncogenic resistance of DPSCs are much better delineated in this new version.

  1. It ultimately remains unclear what the focus/point of this review was. Perhaps it is to set-up the idea that DPSCs need further research because unlike other stem cells which give rise to very malignant cancer types, DPSCs don’t? Or to compare various types of naturally occuring stem cells and the cancers they can give rise to?

 The idea to make this review originated from a brainstorming session in our lab where we discussed the characteristics of stem and cancer stem cells that we work routinely with. We are a multidisciplinary team with research expertise in different types of stem cells and cancer stem cells (CSCs), including those from colorectal cancer and glioma. The intestine and the brain arguably represented the two ends of a scale of endogenous stem cell cycling activity (epithelial stem cells: high proliferation; neural stem cells: low proliferation). When we discussed the typical markers and characteristics of CSCs in brain and gut tumors, we were surprised to realize that practically all of them were shared by tumor-resistant DPSCs, and decided to put in a word about that. So, the idea was first to establish a profile of CSC markers from different tumors, and then challenge that putative CSC marker profile with the findings made in DPSCs. We have tried to justify better our choice of stem and CSCs so the motivation for the work is better framed in this new version.

  1. The Title doesn’t actually reflect the review. It really must be revised to reflect the review as-written more accurately.

We have tentatively modified the title to make it more appealing and to reflect better the aim of this review:

Is there a such thing as a genuine Cancer Stem Cell Marker? Perspectives from the Gut, the Brain and the Dental Pulp

  1. Dental pulp cells are referred to as having “intrinsic characteristics” (line 94) or “other critical aspects” (line 511) or a “tight regulation of physiology” (line 535) that make them resistant to malignant transformation.  However, it is unclear what specific factors, proteins, etc., the authors are precisely referring to, aside from the example of PTEN offered at the end. These instances should be reworded or removed in order to accurately portray potential evidence-based reasons that DPSCs may be resistant to becoming cancerous.

There are very few reports that directly prove the hypothesis that DPSCs are less prone to transformation, except that study of  PTEN (Shen et al. Nature Comm. 2019). However, we have the unquestionable fact that DPSCs do not contribute to malignant neoplasms. We have omitted or reworded all these expressions and tried to stick to the experimental support for every affirmation. We add some valuable information of the expression of tumor suppressor genes by DPSCs, in a context of cell cycle arrest after genotoxic damage. However, regarding which factors and proteins are specifically responsible for the non-propensity of DPSCs to transformation, there is really still very little experimental evidence, and many open questions.

  1. Although it is clear to me why dental pulp stem cells are noteworthy, since they do not seem to give rise to cancers, the authors should validate further why considering specifically brain and gut stem cells are the most logical contrasting choice for comparison. Without further validation of these other tissues’ stem cells, the choice of these two alternatives that do give rise to aggressive cancers seems to simply be a convenient most-severe choice of the authors. Are there tissues that are in many ways very similar to dental pulp in many ways (more similar than gut/brain), but that actually do give rise to cancers (even if not that severe) that would be informative to compare to DPSCs? Perhaps this tissue type would have just a small number of variables that makes the difference in terms of markers, etc.

The choice to focus on tumors of the gut and the brain was to cover the largest possible range of endogenous stem cell activity/proliferation, and of course we also admit it came handy to us that they were both tumors on which we had a previous working experience.

The tissues with the highest resemblance to the dental pulp are mesenchymal-like loose connective stromal tissues that can be found distributed throughout the human body and enriched in some particular places like the skin, the bone marrow, the adipose tissue, and the umbilical cord, among others. All these tissues contain mesenchymal stem cells (MSCs), which share a great deal of markers, but not all of them, with DPSCs. These loose connective tissues can also give rise to soft tissue sarcomas (fibrosarcomas, liposarcomas, mesenchymomas, etc.) with a very low cumulative incidence in the human population. However, some CSC phenotypes have also been identified in these cases, often showing the expression of some of the mentioned markers (CD133, CD44, etc.) in soft tissue sarcomas.

It was however out of the scope of this review to make a systematic comparison between DPSCs and MSCs. This has been done elsewhere and many excellent reviews can be found on the topic. However, we do acknowledge that it is interesting to include information about loose stromal tissues with a high resemblance to the dental pulp, for a closer comparison. We have tried to include relevant references to MSCs and CSCs of their corresponding soft tissue sarcomas, all the while trying not to increase too much the reference list. In doing so, some not-so-relevant sections of the article have necessarily been discarded, to make room for this new information.

  1. Table 1 is extremely confusing. It needs to be completely overhauled to be intelligible. It is hard to know why there is information without citations in some locations. If it is not known, then the “?” symbol is used; if it is known that there is negative expression, that should also be cited, but often it is not. The presence of data but lack of a citation in many places is vexing.

We sincerely apologize for the trouble this table has caused. Now all relevant expressions have been coded according to a heatmap color scale, and all supporting references (including cases of no expression) have been accordingly specified. The table keeps being a bit complex at a first glance, but the presence/absence of expression can be now fast and visually assessed by the help of colors. Furthermore, all the supporting positive and negative expression literature references have been now included, and in case there was no concluding evidence we stuck to yellow color. The reported increases and decreases of expression have also been specified by up/down pointing arrows.

  1. It is hard to tell the “+/–/?” signs from the citations; could the authors use color? How about reformatting the Table to be like a heat map, with red vs. green rectangles for each cell of the Table instead of + and – signs? 

We thank the reviewer for this very valuable suggestion that we fully implement in this new version. We agree that a color heat map scale provides a very nice visual way to observe differences in expression at a quick glance, where green: detectable expression, red: no expression, yellow: unknown or yet to be assessed expression. We still keep +/-signs to show the presence or absence of supporting literature references, as opposed to “?”, which would mean that we found no related references.

  1. Also, the use of “negative/positive expression” is not clear; do the authors mean differentially expressed (up vs. down relative to normal cells or is it detectable vs. undetectable, or…)? If it is relative to normal/wild-type, then how is their normal-tissue example a relevant column if the cancer cells are based on expression level from this column? 

In table1 we include data of qualitative expression (presence/absence, in colors) for each marker and cell type. In some the cases where quantitative differences of expression are reported, those are specified by up/down pointing arrows.

  1. It would also be helpful to add lines between tissue types to easily differentiate the boundary between them. 

Done.

  1. Figure 2 just seems to be a repeat of Table 1.

We agree that the information provided by them was very similar in the previous version. We have modified Figure 2 to a color heatmap where we only concentrate on selected differentially expressed CSC markers, to make it more visually attractive. We complete Figure 2 with data showing the relative incidence of each cancer type.

  1. The layout of Figure 2 is overly complex and hard to quickly understand. For example, if the top row is normal, non-cancerous cells, why not have the word “Normal” or “Wild-type” or something as a label to the side of the chart, and “Cancerous” for the lower row?

Done as suggested.

  1. There is too much redundancy of text; in particular, the cellular marker genes and boldface type are very confusing and it is way too hard to quickly perceive the authors’ points (in fact, I am not sure the points are clear even after scrutinizing the figure). I am sure the authors can apply more critical thinking to this chart and come up with a more streamlined presentation. One simple clarifying suggestion that reduces text and improves clarity would be to, for example, have “Nestin” stated just once on the left of the table and then use a + vs – across rows for each cell type to show if it is present/absent (and the authors may want to use ++ or something if it is thought to be very high). 

The initial idea was to highlight markers that showed qualitative expression changes between stem cells and their corresponding CSCs. However, some mistakes occurred during figure editing so some markers (e.g. MUSASHI) appeared in bold letter in all cases. We sincerely apologize for this.

In this new version, we have decided to show only a selection of CSC markers in a color heatmap scale, so the similarity between CSCs and DPSCs is visually evidenced by a shared green color (showing shared marker expression). 

  1. If boldface indicates differing presencence/absence between tissues, why, e.g., is MUSASHI bold between ISCs and CCSCs yet in both cases it says “MUSASHI+” (in bold)??

This was clearly a mistake of ours. Thank you for remarking it. We have deleted MUSASHI from the new figure version.

  1. In Figure 1, the contents of and differences between the images is not as clear as possible. It would be helpful to add +serum and -serum or something similar on the side, and to add the cell type in addition to the existing labels. It may seem more logical to reverse the order such that the “wild type,” healthy cells are at the top. Finally, scale bars are absent from three of the images.

We have modified the figure including the captions +/-serum accordingly.

Minor concerns: 

  1. The Figure 2 legend has 2 redundant title-like statements. Also, the images used look like they are taken from a textbook or something? Should there be citations in this case; who owns/copyrighted (?) these images or did the authors actually draw them?

The images of Figure 2 are all Free Open Access from Smart Servier Medical Art (www.smart.servier.com). We forgot to introduce credits to the original source in the previous version. We amend this in the new version.

  1. The image quality of Figure 2 is slightly pixelated and would appear cleaner. The clarity of the figure would also be improved if the titles of the cell type were larger than titles of markers.

Amended.

  1. Line 51: How are the presence of gut parasites and/or an increased turnover rate specifically acting as a natural defense mechanism?

It is described in the corresponding reference. An increased epithelial proliferation basically helps to expel the parasites out. Reworded sentence in the new version.

  1. Line 90-91: before 1937 how many cases were reported?

Apart from the case of 1937, back in 1904 there was also another reported case of a primary dental pulp neoplasm, but this was of epithelial origin (epithelioma of the dental pulp), therefore not originating in the dental pulp stroma containing DPSCs.

Otherwise, back in the late nineteenth century it was pretty common to diagnose dental pulp neoplasms as “pulpitis chronic sarcomatosa”. Those were associated to bacterial infections arising from deficient dental care. However closer examinations demonstrated that those cases were not dental pulp sarcomas per se, but a colonization of the exposed dental pulp space by the gingival epithelium.

  Reference: Neuhaus, K. W. (2011). Dental Pulp Neoplasms. Encyclopedia of Cancer, 1084–1086. doi:10.1007/978-3-642-16483-5_1559 

  1. Line 234-240: seems out of place, should be new paragraph, new section, or repositioned into the conclusion when discussing DPSCs

Amended.

  1. Line 248: how did the expression change? Does it increase or decrease? By how much?

Its expression is reported to increase during osteogenic differentiation, showing a similar trend to Runx2 (Padial-Molina et al. 2019).

  1. Line 249: why is DCLK1 interesting? Little is known about it, its role is "controverted" and nothing is known related to DPSCs

Section removed from main text and figures, together with other markers like GPD1.

  1. Line 377/citation 186: cite more primary literature here

Done.

  1. Line 385-386 this is misleading because it implies that the absence of telomerase results in cellular resistance to oncogenic transformation

We have remade that section.

  1. Line 393: What is meant by “consistent”? Consistently high? low? average? compared to what?

By “consistent” we meant stable and easily detectable levels. Reworded.

  1. Line 437-438 therapeutic potential seems out of place, not relating to DPSCs

We have decided to remove the whole former section of “resistance to radiation and chemotherapy” as it was actually not so related to the differential marker expression between CSCs and DPSCs.

  1. Line 442 why is that logical?

Removed from new version.

  1. Line 474 what is meant by “today”? Reference from 2001?

Reworded.

  1. Line 488 what is “solid’ telomerase activity? Same level as something else? Increased relative to something else?

Similar to the “consistent” issue before ( #10). We meant “stable and easily detectable”. Reworded in new version.

  1. Line 501: add PTEN to Table 1/Figure 2

Done.

Spelling and grammar concerns

  1. Line 16:  The conversion of the healthy stem cells

Reworded.

  1. Line 19:  However, this is not the case of the dental pulp

Rewritten.

  1. Line 21: regarding to: remove “to”

Done.

  1. Line 25:  there exist almost literally no reports about

The sentence has been rewritten.

  1. Line 26: reword “This raises the question of what is so special about the dental pulp” to…

The sentence has been reformulated.

  1. Line 48: reword intense

Done.

  1. Line 49: a very few days: remove “very”

Done.

  1. Line 58: takes places: change to “takes place”

Done.

  1. Line 73: Glioblastoma: should not be capitalized

Corrected.

  1. Line 81: this CSCs: should be these CSCs

Done.

  1. Line 126: incomplete sentence

Fixed.

  1. Line 149: is “positivity” is the best way to describe the purported presence of CD133?

We changed “positivity” for “a positive staining”.

  1. Line 259: use of “differentiates” (meaning “is different from”) is confusing because it is in the context of stem cells, which can differentiate in the sense of specifying cell types.

We changed “differentiates” for “distinguish”.

  1. Line 388: to: change to “of”

Done.

  1. Line 409: reposition “from the latter” to immediately follow “gain”

Section removed.

  1. Line 412 noteworthy: add “it is (noteworthy) that…”

Section removed.

  1. There are various sentences/sections which need to be reworked because they are repetitive.

Done.

  1. Proofreading is also needed to address some grammar/wording choices

Done.

  1. In general, more transition sentences are needed to better connect the flow of the paper.

We have tried to improve this aspect. Please check.

Reviewer 2 Report

This is an excellent and very timely review about differences in transforming from tissue to cancer stem cells using 3 different types of stem cells to compare between. The review is written in a very good English with only very occasional minor grammar mistakes. 

However, there are a few issues which have to be addressed.

1. In line 36 and others the authors write that adult tissue stem cells renew postmitotic cell populations which is only partially true for some cell types such as neurons while most other cell types reviewed here are mitotic and the authors even write about "cell turnover". Both, intestinal and glial cells are highly MITOTIC. Please correct this statement.

2. In lines 53 and 54 the authors write about differentiated cells, neurons and glia cells, having self-renewal abilities. This is incorrect and just refers to their stem/progenitor cells. 

3. Several statements (for example, lines 76, 86, 507 etc) lack appropriate references.

4. line 98: does "feats" mean "features"?

5. Fig. 1: the differentiated cells in the lower rows should be better described and also have scale bars.

6. Lines 173/4 While the authors have stated before that there is no LGR5 in mice, just in humans, they now talk about no effects in LGR5 knock-out mice. How can this be if mice don't have the gene?

7. line 267: the authors write about non-pluripotent neural progenitor cells. However, ALL adult tissue stem cells are NON-pluripotent! Only ESCs are pluri-potent! Please correct this mistake.

8. Table heading should be above the table.

9. Importantly, the chapter about telomerase activity (TA) is incomplete. Except for some general statements only dental SC are described for their TA while nothing is described for intestinal and neural SC which both have TA and hTERT expression (see for example Schepers et al., 2010 and Beck et al., 2011, Miura et al., 2001 Neural stem cells lose telomerase activity upon differentiating into astrocytes and many more.

This also has to be corrected for these 2 stem cell types in figure 2 and is therefore not different for the 3 SC types.

10. In contrast, human mesenschymal stem cells (MSC) are almost the only human adult tissue stem cell type that does not have any meaningful telomerase activity while you claim that they have.

11. Importantly, telomerase activity is inducible in most adult stem cells when they are activated and not quiescent any more. Thus, they are potentially positive for TA (Hiyama and Hiyama, 2007).

12. Finally, did you consider unusually strong tumour suppressor genes in dental pulp SC as a possible reason for their resistance to transform into CSC?

13. Please explain all abbreviations when first used in the text, for example BMMSC etc.

Author Response

  1. This is an excellent and very timely review about differences in transforming from tissue to cancer stem cells using 3 different types of stem cells to compare between. The review is written in a very good English with only very occasional minor grammar mistakes. 

We sincerely appreciate your very valuable and pertinent remarks, together with your overall positive opinion on this manuscript.

However, there are a few issues which have to be addressed.

  1. In line 36 and others the authors write that adult tissue stem cells renew postmitotic cell populations which is only partially true for some cell types such as neurons while most other cell types reviewed here are mitotic and the authors even write about "cell turnover". Both, intestinal and glial cells are highly MITOTIC. Please correct this statement.

Thank you for your remark. We remade this sentence in order to avoid confusion to the readers.

  1. In lines 53 and 54 the authors write about differentiated cells, neurons and glia cells, having self-renewal abilities. This is incorrect and just refers to their stem/progenitor cells. 

We reformulated the sentence. Please check.

  1. Several statements (for example, lines 76, 86, 507 etc) lack appropriate references.

We added the references to their respective statements as follows:

The Cancer Stem Cell theory states that tumor growth is fueled by small numbers of tumor stem cells (CSCs) hidden within the bulk of the tumor mass [15]. Much as normal cell renewal in healthy adult tissues depends on activation and proliferation of their endogenous stem cells, cell renewal in malignant tumors would depend on the activation of CSCs [16]. This theory explains clinical observations, such as the recurrence of tumors after initially successful therapy, and the phenomena of tumor dormancy and metastasis [17].

References:

  1. Visvader, J.E.; Lindeman, G.J. Cancer stem cells in solid tumours: accumulating evidence and unresolved questions. Nat Rev Cancer 2008, 8, 755–768, doi:10.1038/nrc2499.
  2. Sell, S. Stem cell origin of cancer and differentiation therapy. Crit Rev Oncol Hematol 2004, 51, 1–28, doi:10.1016/j.critrevonc.2004.04.007.
  3. Wang, K.; Wu, X.; Wang, J.; Huang, J. Cancer stem cell theory: therapeutic implications for nanomedicine. Int J Nanomedicine 2013, 8, 899–908, doi:10.2147/IJN.S38641.
  4. line 98: does "feats" mean "features"?

We reformulated that section

  1. Fig. 1: the differentiated cells in the lower rows should be better described and also have scale bars.

The scale bars are the same of the upper images. We improved the figure.

  1. Lines 173/4 While the authors have stated before that there is no LGR5 in mice, just in humans, they now talk about no effects in LGR5 knock-out mice. How can this be if mice don't have the gene?

We were talking just about murine NSCs, not ISCs. We have reformulated this whole section to avoid confusion. Please check it.

  1. line 267: the authors write about non-pluripotent neural progenitor cells. However, ALL adult tissue stem cells are NON-pluripotent! Only ESCs are pluri-potent! Please correct this mistake.

Of course, we did not mean that adult tissue stem cells were pluripotent. We corrected the sentence (line 325) as follows:

Interestingly, neural progenitor cells have also been reported to express mRNA for NANOG and OCT4 [143]

  1. Table heading should be above the table.

We moved the table heading to the upper position.

  1. Importantly, the chapter about telomerase activity (TA) is incomplete. Except for some general statements only dental SC are described for their TA while nothing is described for intestinal and neural SC which both have TA and hTERT expression (see for example Schepers et al., 2010 and Beck et al., 2011, Miura et al., 2001 Neural stem cells lose telomerase activity upon differentiating into astrocytes and many more.

This also has to be corrected for these 2 stem cell types in figure 2 and is therefore not different for the 3 SC types.

Thank you very much for this very pertinent and meaningful comment. We apologize for these mistakes, and we correct this manuscript section and Figure 2 accordingly.

At the line 434 we added the sentence: “Interestingly, it has been reported that a mutated TERT fragment is able to induce brain cancer stemness independently of it telomerase activity function [203] [205].”

At the line 441 we added the sentence: “However, multipotent stem cells such as ISCs, MSCs, DPSCs and NSCs all present a basal telomerase activity and hTERT expression [209–212].”

At the line 443 we added the following paragraph: “Brain telomerase activity in adult mice has been found to be restricted to the subventricular zone and olfactory bulb [213]. It plays an important role in cell proliferation in the adult but not in embryonic NSCs [214]. Telomere length has also been demonstrated to be important for neuronal differentiation and neuritogenesis [215] (see also review [216]). Its deficiency leads to a compromised olfactory bulb neurogenesis [214] although NSCs lose telomerase activity upon differentiation into astrocytes [217]. DPSCs also lose progressively their telomerase activity upon their spontaneous in vitro differentiation to osteoblastic/odontoblastic cells in conditions of high culture passages [211].

References:

  1. Beck, S.; Jin, X.; Sohn, Y.-W.; Kim, J.-K.; Kim, S.-H.; Yin, J.; Pian, X.; Kim, S.-C.; Nam, D.-H.; Choi, Y.-J.; et al. Telomerase Activity-Independent Function of TERT Allows Glioma Cells to Attain Cancer Stem Cell Characteristics by Inducing EGFR Expression. Mol Cells 2011, 31, 9–15, doi:10.1007/s10059-011-0008-8.
  2. Schepers, A.G.; Vries, R.; van den Born, M.; van de Wetering, M.; Clevers, H. Lgr5 intestinal stem cells have high telomerase activity and randomly segregate their chromosomes. EMBO J 2011, 30, 1104–1109, doi:10.1038/emboj.2011.26.
  3. Ninagawa, N.; Murakami, R.; Isobe, E.; Tanaka, Y.; Nakagawa, H.; Torihashi, S. Mesenchymal stem cells originating from ES cells show high telomerase activity and therapeutic benefits. Differentiation 2011, 82, 153–164, doi:10.1016/j.diff.2011.07.001.
  4. Horibe, H.; Murakami, M.; Iohara, K.; Hayashi, Y.; Takeuchi, N.; Takei, Y.; Kurita, K.; Nakashima, M. Isolation of a stable subpopulation of mobilized dental pulp stem cells (MDPSCs) with high proliferation, migration, and regeneration potential is independent of age. PLoS ONE 2014, 9, e98553, doi:10.1371/journal.pone.0098553.
  5. Jeon, B.-G.; Kang, E.-J.; Kumar, B.M.; Maeng, G.-H.; Ock, S.-A.; Kwack, D.-O.; Park, B.-W.; Rho, G.-J. Comparative analysis of telomere length, telomerase and reverse transcriptase activity in human dental stem cells. Cell Transplant 2011, 20, 1693–1705, doi:10.3727/096368911X565001.
  6. Caporaso, G.L.; Lim, D.A.; Alvarez-Buylla, A.; Chao, M.V. Telomerase activity in the subventricular zone of adult mice. Mol Cell Neurosci 2003, 23, 693–702, doi:10.1016/s1044-7431(03)00103-9.
  7. Ferrón, S.; Mira, H.; Franco, S.; Cano-Jaimez, M.; Bellmunt, E.; Ramírez, C.; Fariñas, I.; Blasco, M.A. Telomere shortening and chromosomal instability abrogates proliferation of adult but not embryonic neural stem cells. Development 2004, 131, 4059–4070, doi:10.1242/dev.01215.
  8. Ferrón, S.R.; Marqués-Torrejón, M.A.; Mira, H.; Flores, I.; Taylor, K.; Blasco, M.A.; Fariñas, I. Telomere shortening in neural stem cells disrupts neuronal differentiation and neuritogenesis. J Neurosci 2009, 29, 14394–14407, doi:10.1523/JNEUROSCI.3836-09.2009.
  9. Liu, M.-Y.; Nemes, A.; Zhou, Q.-G. The Emerging Roles for Telomerase in the Central Nervous System. Front Mol Neurosci 2018, 11, 160, doi:10.3389/fnmol.2018.00160.
  10. Miura, T.; Katakura, Y.; Yamamoto, K.; Uehara, N.; Tsuchiya, T.; Kim, E.H.; Shirahata, S. Neural stem cells lose telomerase activity upon differentiating into astrocytes. Cytotechnology 2001, 36, 137–144, doi:10.1023/A:1014016315003

  1. 10 In contrast, human mesenschymal stem cells (MSC) are almost the only human adult tissue stem cell type that does not have any meaningful telomerase activity while you claim that they have.

We are sorry for the confusion, we clarified the message by adding new references and the following sentence at the line 459: “However, this mechanism is very active in malignant tumors of mesenchymal origin [225,226].”

References:

  1. Lawlor, R.T.; Veronese, N.; Pea, A.; Nottegar, A.; Smith, L.; Pilati, C.; Demurtas, J.; Fassan, M.; Cheng, L.; Luchini, C. Alternative lengthening of telomeres (ALT) influences survival in soft tissue sarcomas: a systematic review with meta-analysis. BMC Cancer 2019, 19, 232, doi:10.1186/s12885-019-5424-8.
  2. Venturini, L.; Motta, R.; Gronchi, A.; Daidone, M.; Zaffaroni, N. Prognostic relevance of ALT-associated markers in liposarcoma: a comparative analysis. BMC Cancer 2010, 10, 254, doi:10.1186/1471-2407-10-254.
  3. Importantly, telomerase activity is inducible in most adult stem cells when they are activated and not quiescent any more. Thus, they are potentially positive for TA (Hiyama and Hiyama, 2007).

We agree that TERT presence /absence of expression does  not really constitute a differential factor to distinguish between normal and cancer stem cells after all.

Thank you very much for pointing this out.

  1. Finally, did you consider unusually strong tumour suppressor genes in dental pulp SC as a possible reason for their resistance to transform into CSC?

We are sincerely grateful for this meaningful suggestion. We have remade the last section of the article to highlight the fact that DPSCs have the capability to activate strong p53/p21 dependent DNA-damage responses. We link these evidences with the discussion about PTEN and other reported tumor suppressors like let-7c microRNAs. We think this information contributes to delineate much better some of the possible mechanisms involved in the oncogenic resistance of DPSCs.

  1. Please explain all abbreviations when first used in the text, for example BMMSC etc.

Done.

Reviewer 3 Report

This review aims to compare various cancer stem cells to dental pulp stem cells. The idea is original and of interest because of the intriguing resistance to oncogenesis of this tissue. The manuscript is well written and offers a good overview of the issue. I believe that this review will be useful for anyone studying the relationship between stem cells and cancer.

Please carefully check all the references: for example ref #100 is not appropriate 

Author Response

This review aims to compare various cancer stem cells to dental pulp stem cells. The idea is original and of interest because of the intriguing resistance to oncogenesis of this tissue. The manuscript is well written and offers a good overview of the issue. I believe that this review will be useful for anyone studying the relationship between stem cells and cancer.

We are sincerely grateful for your kind consideration of this article. After the revision round, we have taken steps to tune some of our arguments and avoid some confusions/mistakes. We hope you will like this new version even better.

Please carefully check all the references: for example ref #100 is not appropriate 

We have performed a new careful revision of the reference field. We apologize for these mistakes as they were really a lot of references to introduce. We hope everything to be OK now.